

# High fat-fed GPR55 null mice display impaired glucose tolerance without concomitant changes in energy balance or insulin sensitivity but are less responsive to the effects of the cannabinoids rimonabant or Δ(9)-tetrahydrocannabivarin on weight gain

Edward T. Wargent[1], Malgorzata Kepczynska[2], Mohamed Sghaier Zaibi[1], David C. Hislop[2], Jonathan R.S. Arch[1] and Claire J. Stocker[2]

[1] Institute of Translational Medicine, University of Buckingham, Buckingham, United Kingdom
[2] Medical School, University of Buckingham, Buckingham, United Kingdom

Corresponding author
Claire J. Stocker,
claire.stocker@buckingham.ac.uk

## ABSTRACT

**Background**. The insulin-sensitizing phytocannabinoid, Δ(9)-tetrahydrocannabivarin (THCV) can signal partly via G-protein coupled receptor-55 (GPR55 behaving as either an agonist or an antagonist depending on the assay). The cannabinoid receptor type 1 (CB1R) inverse agonist rimonabant is also a GPR55 agonist under some conditions. Previous studies have shown varied effects of deletion of GPR55 on energy balance and glucose homeostasis in mice. The contribution of signalling via GPR55 to the metabolic effects of THCV and rimonabant has been little studied.

**Methods**. In a preliminary experiment, energy balance and glucose homeostasis were studied in GPR55 knockout and wild-type mice fed on both standard chow (to 20 weeks of age) and high fat diets (from 6 to 15 weeks of age). In the main experiment, all mice were fed on the high fat diet (from 6 to 14 weeks of age). In addition to replicating the preliminary experiment, the effects of once daily administration of THCV (15 mg kg$^{-1}$ po) and rimonabant (10 mg kg$^{-1}$ po) were compared in the two genotypes.

**Results**. There was no effect of genotype on absolute body weight or weight gain, body composition measured by either dual-energy X-ray absorptiometry or Nuclear Magnetic Resonance (NMR), fat pad weights, food intake, energy expenditure, locomotor activity, glucose tolerance or insulin tolerance in mice fed on chow. When the mice were fed a high fat diet, there was again no effect of genotype on these various aspects of energy balance. However, in both experiments, glucose tolerance was worse in the knockout than the wild-type mice. Genotype did not affect insulin tolerance in either experiment. Weight loss in rimonabant- and THCV-treated mice was lower in knockout than in wild-type mice, but surprisingly there was no detectable effect of genotype on the effects of the drugs on any aspect of glucose homeostasis after taking into account the effect of genotype in vehicle-treated mice.

**Conclusions**. Our two experiments differ from those reported by others in finding impaired glucose tolerance in GPR55 knockout mice in the absence of any effect on body weight, body composition, locomotor activity or energy expenditure. Nor could we detect any effect of genotype on insulin tolerance, so the possibility that GPR55 regulates glucose-stimulated insulin secretion merits further investigation. By contrast with the genotype effect in untreated mice, we found that THCV and rimonabant reduced weight gain, and this effect was in part mediated by GPR55.

# INTRODUCTION

Previous work from our laboratory has shown that the plant-derived cannabinoid, $\Delta(9)$-tetrahydrocannabivarin (THCV) improves insulin sensitivity in diet-induced obese and ob/ob mice (*Wargent et al., 2013a*; *Wargent et al., 2013b*) but it is not clear which receptor or receptors mediate its effects. Cannabinoids, signal partly via GPR55 (*Pertwee, 2007*; *Sharir & Abood, 2010*). THCV was a high efficacy, low affinity agonist of ERK1/2 phosphorylation when hGPR55 was expressed in HEK293 cells but it inhibited L-$\alpha$-lysophosphatidylinositol signalling (*Anavi-Goffer et al., 2012*). The cannabinoid receptor type 1 (CB1R) inverse agonist rimonabant, which was in the past used for the treatment of obesity, is also a GPR55 agonist (*Kapur et al., 2009*; *Henstridge et al., 2010*), although under some conditions it can behave as an antagonist (*Lauckner et al., 2008*; *Anavi-Goffer et al., 2012*).

There is conflicting evidence as to whether GPR55 agonists or antagonists might be of benefit in the treatment of obesity or type 2 diabetes (*Lipina et al., 2012*; *Moreno-Navarrete et al., 2012*; *Henstridge, Brown & Waldhoer, 2016*). Some findings in humans suggest that GPR55 receptor *antagonists* should reduce food intake and body weight (*Henstridge, Brown & Waldhoer, 2016*). By contrast, in support of GPR55 *agonists*, two studies found that GPR55 knockout mice showed increased adiposity and insulin resistance associated with decreased locomotor activity (*Meadows et al., 2016*; *Lipina et al., 2019*), although another failed to demonstrate increased adiposity and locomotor activity was actually increased during the first six hours of the dark period (*Bjursell et al., 2016*). Further support for the potential of GPR55 agonists in the treatment of type 2 diabetes comes from two studies that have found that the GPR55 agonist O-1602 stimulated insulin secretion from wild-type but not GPR55 -/- murine islets of Langerhans (*Romero-Zerbo et al., 2011*; *Liu et al., 2016*). In one of these studies (*Romero-Zerbo et al., 2011*), it was also shown that O-1602 stimulated insulin secretion and improved glucose tolerance in vivo in rats.

Previous studies in GPR55 knockout mice have mostly been conducted using mice fed on a standard chow diet. Here we first compared GPR55 knockout and wild-type mice fed on both standard chow and high fat diets. We found that oral glucose tolerance was worse in GPR55 knockout than in wild-type mice when the mice were fed on a high fat diet but not when they are fed on a chow diet. To investigate the role of GPR55 in responses to

THCV and rimonabant, we therefore compared metabolic responses to these drugs in wild type and GPR55 knockout mice fed on a high fat diet. We report that THCV and especially rimonabant had less effect on body weight gain in GPR55 knockout than in wild-type mice but we were unable to demonstrate genotype influenced changes in glucose homeostasis in response to THCV or rimonabant.

## MATERIALS AND METHODS

### Mice

Two male and five female GPR55$^{+/-}$ mice on a C57Bl/6 background were kindly supplied, with the permission of AstraZeneca, Macclesfield, UK, by Professor Cherry Wainwright of the Institute for Health and Welfare Research, The Robert Gordon University Aberdeen AB10 1FR, UK. They were bred to produce GPR55$^{-/-}$ ('knockout') and wild-type mice. The final breeding-round for the current studies was between homozygous wild-type or knockout mice.

In the preliminary experiment, the purpose of which was solely to compare the phenotypes of wild-type and GPR55 knockout mice, the intention was to use 12 wild-type and 12 knockout male mice in each experiment (chow-fed or high fat diet-fed), housed in pairs. Only 11 knockout mice were available for the high fat diet experiment, however.

In the main experiment, which focussed on mice fed on a high fat diet and investigated whether responses to THCV (15 mg kg$^{-1}$ po once daily; GW Research Ltd., Cambridge, UK) and rimonabant (10 mg kg$^{-1}$ po once daily) differed between genotypes, the intention was again to use 12 wild-type and 12 knockout mice, housed in pairs, in each group. However, one mouse died before the experimental period began, one mouse died in each of the control groups, and one mouse died in the rimonabant wild-type group (tumour found in chest). All data for these mice have been excluded. The vehicle for both THCV and rimonabant was 2.5% ethanol in sesame seed oil (10 ml kg$^{-1}$)l.

The mice were housed at 24–26 °C with lights on at 08:00 and off at 20:00. Mice were housed in pairs in the preliminary study and 3 per cage in the main study. They were fed at weaning on chow (Beekay rat and mouse diet No 1; BK001E; Beekay Feed, B&K Universal Limited) and from six weeks of age on a high fat diet (metabolizable energy: 60% fat; 20% carbohydrate; 20% protein; Research Diets, New Brunswick, NJ, USA; product # D12492). Food and water were provided *ad libitum*. Cages had solid bases with sawdust for foraging and digging. Cotton fibre nestlets and Enviro-Dri paper were provided as bedding and cover. Cardboard houses and tunnels were also provided for shelter, exploration, and gnawing. Wood chew sticks were also provided for gnawing. Interlocking PVC sections were used for climbing and compartmentalisation.

All procedures involving animals were conducted in accordance with the University of Buckingham Animals (Scientific Procedures) Act 1986 (ASPA) Ethical Review Board and theASPA Project Licence (PPL) licence number: 70/7164. ARRIVE guidelines were followed in the reporting of the experiments. Mice were inspected daily for adverse effects and after every procedure. At the end of the study all mice were killed by concussion of the brain by striking the cranium followed by cervical dislocation.

Criteria for euthanizing animals before the end of study were as follows. Any animal showing signs of mis-dosing or damage after oral gavage, such as by coughing/choking or collapsing after administration of substances will be killed by an approved method. Animals receiving intraperitoneal dosing will be observed immediately after dosing and if treatment is for a prolonged period, we will monitor the animals for signs of pain and distress that may indicate peritonitis (hunching, subdued behaviour, hind limb extension). Any animal showing signs of damage will be humanely killed. Following blood sampling animals showing lasting signs of damage or exceeding the mild severity limit will be humanely killed. In the insulin sensitivity tests any animal that shows signs of torpor after insulin administration will immediately be given glucose and/or glucagon by the intra-peritoneal route, monitored continuously and killed if it fails to respond to stimulation or does not recover within 20 min. None of these criteria was necessary to be implemented in the preliminary study. Four animals in the main study experienced distress due to oral mis-dosing and were culled by concussion followed by cervical dislocation. This incidence rate was within the expected maximum of less than 1 in a thousand doses set out in the terms of the project licence.

## EXPERIMENTAL METHODS

Energy expenditure was measured by open circuit indirect calorimetry with mice in their home cages (*Arch et al., 2006*).

For the measurement of locomotor activity, mice were kept in the cages (28 × 12 cm) in which they normally housed. They remained in their pairs. Video camera shots were taken every hour for 10 min, beginning 1 h before the dark period. Thus, the first recording was at 19:00 and the last at 08:00. The recordings were digitally divided by black lines into three equal rectangles after filming. Horizontal locomotor activity was assessed by one independent observer from the number of times a mouse crossed a line during those 10 min in a blinded study.

Body fat and lean content was measured using a Minispec LF90$_{II}$ Nuclear Magnetic Resonance (Bruker Corporation, Germany). Dual-energy X-ray absorptiometry (DXA or DEXA) was also used in the preliminary experiment because it gives a measure of bone density, and it has been reported that bone structure is altered in GPR55 knockout mice (*Whyte et al., 2009*).

Insulin tolerance was measured after mice had been fasted for five hours before being dosed with insulin (Actrapid$^{TM}$ (Centaur Services, UK) at 0.5 units/kg body weight, i.p. for chow-fed mice and 0.75 units/kg bodyweight for high fat diet-fed mice). Blood samples for glucose measurement were taken 10 min and immediately before, and 10, 20, 30, 45 and 60 min after the administration of insulin.

Pancreatic insulin concentration and content were measured at termination. The mice were fasted from 09:00 and humanely killed at 14:00. Pancreatic insulin content (*Wang, Cawthorne & Clapham, 2002*), liver glycogen (*Pearce et al., 2004*), blood glucose and insulin, liver triglycerides and oral glucose and intraperitoneal insulin tolerance tests (*Wargent et al., 2013a*; *Wargent et al., 2013b*) were conducted as described previously.

## Statistics

Results given in the text, and data-points in the figures are shown as the mean $\pm$ SEM. Sample size was calculated by the resource equation method (*Festing & Altman, 2002*). All data sets passed the Anderon-Darling test for normality of distribution (alpha of 0.05). The statistical significance of any differences between vehicle-treated animals and drug-treated animals was determined using Student's $t$-test, or where there were multiple treatments or time-points, 1-way or 2-way ANOVA followed by False Discovery Rate post-tests (FDR; two-stage linear step-up procedure of Benjamini, Krieger and Yekutieli), using Prism 7. ROUT was used to analyse data sets for outliers with no outliers being identified. In the FDR test $Q$ (adjusted $P$ value) is the lowest value that gave 'Yes' in the Discovery column of the analysis. Statistical significance is shown as: $*P$ or $Q < 0.05$, $**P$ or $Q < 0.01$; $***P$ or $Q < 0.001$; $****P$ or $Q < 0.0001$.

# RESULTS

## Preliminary experiment

The mice were fed on chow and then some were fed on a high fat diet from 6 weeks of age.

Weight gain was significantly greater ($Q < 0.0001$ for both wildtype and knockout mice) between 6 and 11 weeks of age in mice fed on a high fat diet (wild type: 12.6 $\pm$ 0.9 g, $n = 12$; knockout:12.2 $\pm$ 1.3 g, $n = 11$) than in mice fed on chow (wild type: 6.1 $\pm$ 0.4 g, $n = 12$; knockout: 5.2 $\pm$ 0.4 g, $n = 12$). However, irrespective of diet, there was no statistically significant effect of genotype on absolute body weight or weight gain (Fig. 1), body composition measured by either DXA or NMR, fat pad weights, food intake, energy expenditure or locomotor activity (Table 1 for chow-fed mice; Table 2 for high fat-fed mice); nor on the time courses of energy expenditure and locomotor activity (results not shown, but are in the Data S1).

In the chow-fed mice, two-way ANOVA with time matching showed no effect of genotype on blood glucose in the glucose tolerance test (Fig. 2A). There was no effect of genotype on plasma insulin at +30 or −30 min relative to the administration of glucose (Fig. 2B). Insulin tolerance, whether expressed in terms of absolute blood glucose levels or the fall in blood glucose following injection of insulin, was no different between genotypes in the chow-fed mice at 20 weeks of age in either absolute blood glucose concentrations (Fig. 2C) or in change in blood glucose (Fig. 2D).

By contrast with the chow-fed mice, there were clear effects of genotype on blood glucose and plasma insulin in the glucose tolerance test and on insulin tolerance in the mice fed on the high fat diet (Fig. 3). Thus, two-way ANOVA with time-matching followed by the FDR test showed that blood glucose was higher in the knockout than the wild-type mice 30 and 60 min after dosing with glucose ($Q < 0.01$; Fig. 3A), and two-way ANOVA followed by the FDR test showed that plasma insulin was also higher in the knockout than the wild-type mice 30 min after administration of glucose ($Q < 0.01$; Fig. 3B).

Blood glucose 20–60 min after administration of insulin in the insulin tolerance test was higher in the knockout mice ($Q < 0.01$; Fig. 3C). However, the fall in blood glucose following the injection of insulin was not significantly different between genotypes (Fig. 3D).

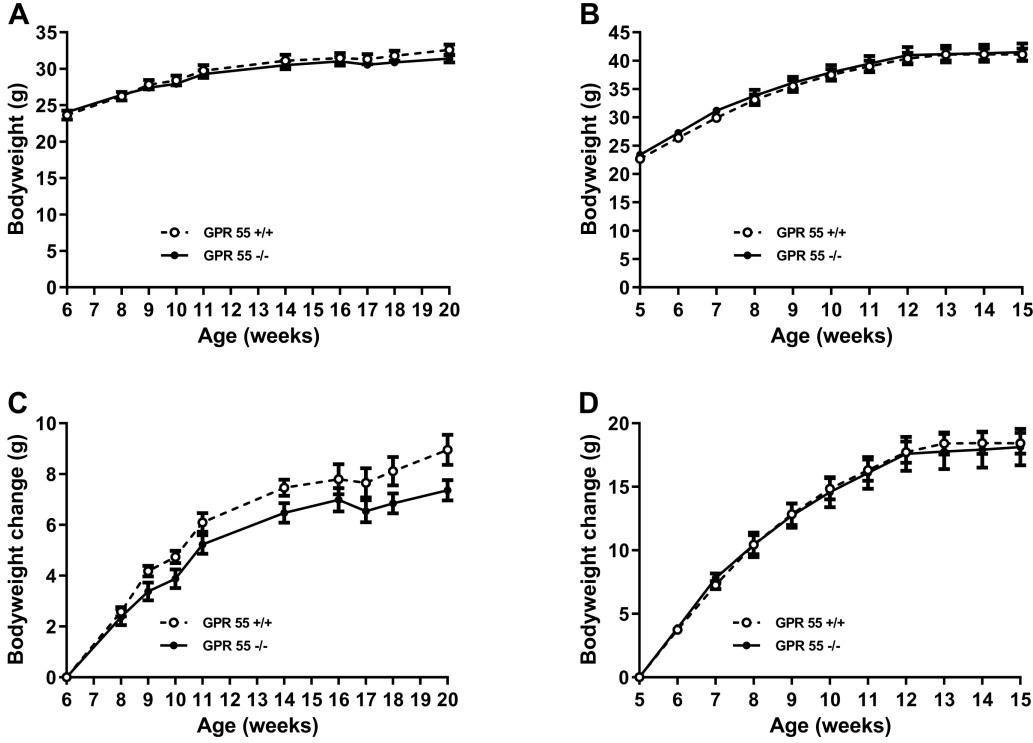

**Figure 1** **Growth trajectory in GPR55 knockout and wild-type mice.** Bodyweight is shown for GPR55 knockout and control mice fed on chow (A), and high fat diet (B). Body weight change is shown for mice fed chow (C) or high fat diet (D). Two-way ANOVA showed no statistically significant differences between wild-type and knockout mice.

**Table 1** **Body composition, energy balance and locomotor activity in chow-fed mice in preliminary experiment.** Bodyweight is shown for GPR55 knockout and control mice fed on chow (A), and high fat diet (B). Body weight change is shown for mice fed chow (C) or high fat diet (D). Two-way ANOVA showed no statistically significant differences between wild-type and knockout mice.

|  | *n* | Wild-type mice | Knockout mice |
|---|---|---|---|
| Body fat content (g) | 12 | $13.01 \pm 0.57$ | $13.13 \pm 0.54$ |
| Body lean content (g) | 12 | $18.02 \pm 0.68$ | $18.63 \pm 0.85$ |
| Epididymal fat pads (g) | 12 | $1.09 \pm 0.09$ | $0.85 \pm 0.10$ |
| Inguinal fat pads (g) | 12 | $0.60 \pm 0.06$ | $0.52 \pm 0.07$ |
| Interscapular fat pads (g) | 12 | $0.108 \pm 0.007$ | $0.118 \pm 0.013$ |
| Food intake over 35 days (g) | 6 | $111 \pm 2$ | $115 \pm 4$ |
| Energy expenditure (kJ/h) | 6 | $3.17 \pm 0.13$ | $3.38 \pm 0.21$ |
| Activity (Line breaks per 10 min) | 6 | $31.3 \pm 1.2$ | $28.8 \pm 1.79$ |

Liver weight, liver glycogen and lipid contents, pancreatic insulin concentration and pancreatic total insulin content were not affected by genotype irrespective of diet (results not shown, but are in the Data S1).

**Table 2   Body composition, energy balance and locomotor activity in high fat-fed mice in preliminary experiment.** Food intake was measured daily when the mice were between 6 and 15 weeks of age, body composition by DEXA and NMR when they were 14 and 15 weeks old respectively, energy expenditure when they were 10 to 11 weeks old, and locomotor activity when they were 13 weeks old. Fat pad weights were measured at termination (15 weeks old). Energy expenditure was measured over 21 h beginning at 14:00 h. Locomotor activity was measured from 19:00 to 08:00 h. Lights were out from 20:00 to 08:00 h. Where $n = 6$ measurements were recorded for pairs of mice and divided by two, except for the knock-out mice that was housed singly. There were no statistically significant differences between wild-type and knockout mice by Student's $t$ test. The lowest value of $P$ was 0.053 for interscapular fat pad weights. Other values were > 0.1.

|  | $n$ | Wild-type mice ($n = 12/6$) | Knockout mice ($n = 12/6$) |
|---|---|---|---|
| Body fat content by DEXA (g) | 12/11 | $21.75 \pm 0.48$ | $23.24 \pm 0.80$ |
| Body lean content by DEXA (g) | 12/11 | $19.37 \pm 0.71$ | $17.91 \pm 0.92$ |
| Bone mineral density (g/cm$^3$) | 12/11 | $0.0531 \pm 0.0007$ | $0.0518 \pm 0.0013$ |
| Body fat content by NMR (g) | 12/11 | $22.89 \pm 0.68$ | $23.17 \pm 0.81$ |
| Body lean content by NMR (g) | 12/11 | $18.18 \pm 0.43$ | $18.33 \pm 0.85$ |
| Epididymal fat pads (g) | 12/11 | $2.18 \pm 0.09$ | $2.05 \pm 0.09$ |
| Inguinal fat pads (g) | 12/11 | $1.712 \pm 0.126$ | $1.513 \pm 0.07$ |
| Interscapular fat pads (g) | 12/11 | $0.108 \pm 0.007$ | $0.118 \pm 0.133$ |
| Food intake over 62 days (g) | 6 | $166 \pm 6$ | $171 \pm 5$ |
| Energy expenditure (kJ/h) | 6 | $3.52 \pm 0.12$ | $3.81 \pm 0.25$ |
| Activity (Line breaks per 10 min) | 6 | $26.6 \pm 4.5$ | $20.96 \pm 5.79$ |

# MAIN EXPERIMENT

The effects of 15 mg kg$^{-1}$ po THCV and 10 mg kg$^{-1}$ po rimonabant were compared between wild-type and GPR55 knockout mice fed on the high fat diet.

## Confirmation of results of preliminary experiment

It was confirmed (for mice fed on a high fat diet) that energy balance is no different between GPR55 knockout and wild-type mice. The mean body weight of the vehicle-treated knockout mice was less than that of the vehicle-treated wild-type mice from the beginning of the study, but differences in body weight were not statistically significant at this or any other time. Analysis of body weight change showed higher increases in body weight for the knockout mice at 21, 28 and 35 days, but by 56 days the wild-type mice had the higher increase (Fig. 4A) and there was no overall effect on genotype on body weight gain. As in the preliminary study, genotype had no effect on food intake (results not shown, but are in the Data S1), energy expenditure (Fig. 5), body composition measurements (Fig. 6) or fat pad weights (Table 3).

It was confirmed that glucose homeostasis is deficient in GPR55 knockout mice. As in the untreated mice in the preliminary study, blood glucose was higher in the vehicle-treated knockout than in the vehicle-treated wild-type mice 30 min ($Q < 0.01$) and 60 min ($Q < 0.05$) after giving glucose in the glucose tolerance test on day 21 (Fig. 7A) and plasma insulin 30 min after giving glucose was higher in the vehicle-treated knockout mice than in the vehicle-treated wild-type mice ($Q < 0.05$; Fig. 8A).

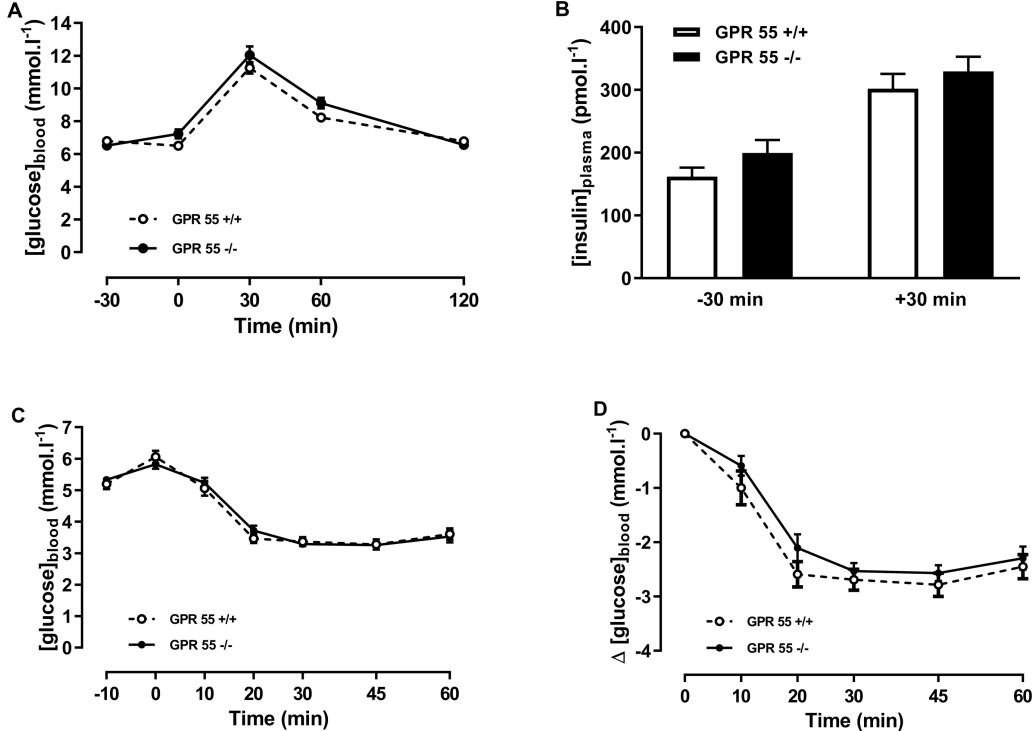

**Figure 2** **Glucose and insulin tolerance in GPR55 knockout and wild-type mice fed on a standard chow diet.** Blood glucose concentration (A) and plasma insulin (B) before and after a glucose load ($t = 0$ min) at age 16 weeks after a 5 h fast, and blood glucose concentration expressed as absolute values (C) or change from $t = 0$ min (D) following an insulin load ($t = 0$ min) at age 20 weeks after a 5 h fast. Two-way ANOVA showed no effect of genotype on blood glucose in the tolerance test, although Fisher's LSD test suggested that blood glucose was higher in the knockout mice at 60 min ($P < 0.05$). There was no statistically significant difference by two-way ANOVA in plasma insulin at $t = -30$ or $t = +30$ in the OGTT. Nor was there any difference between genotypes in blood glucose levels following an insulin load.

Although the same trend was seen as in the preliminary experiment, there was no significant difference in the insulin tolerance curves on day 38 between the vehicle-treated knockout mice and the vehicle-treated wild-type mice (Figs. 9A and 9B). As in the preliminary experiment, there was also no difference in the fall in blood glucose when the data were normalised to the 0 min blood glucose values (Figs. 9C and 9D) or the −10 min values (data not shown, but are in the Data S1).

## Responses to THCV and rimonabant: energy balance

Two-way ANOVA with time-matching showed an overall effect of genotype on body weight gain in response to both THCV ($P < 0.001$; Fig. 4B) and rimonabant ($P < 0.001$; Fig. 4C) from day 8 due to both THCV and rimonabant being less effective in the knockout mice. Although the effect of genotype on body weight gain in THCV-treated mice was statistically significant on days prior to day 56 and over all days, it was not statistically significant for THCV on the final day (day 56), being only 19% less in knockout than in wild-type mice (6.7 g different from control in wild-type mice; 5.5 g different from controls in knockout mice). By contrast, the effect of rimonabant on body weight gain on
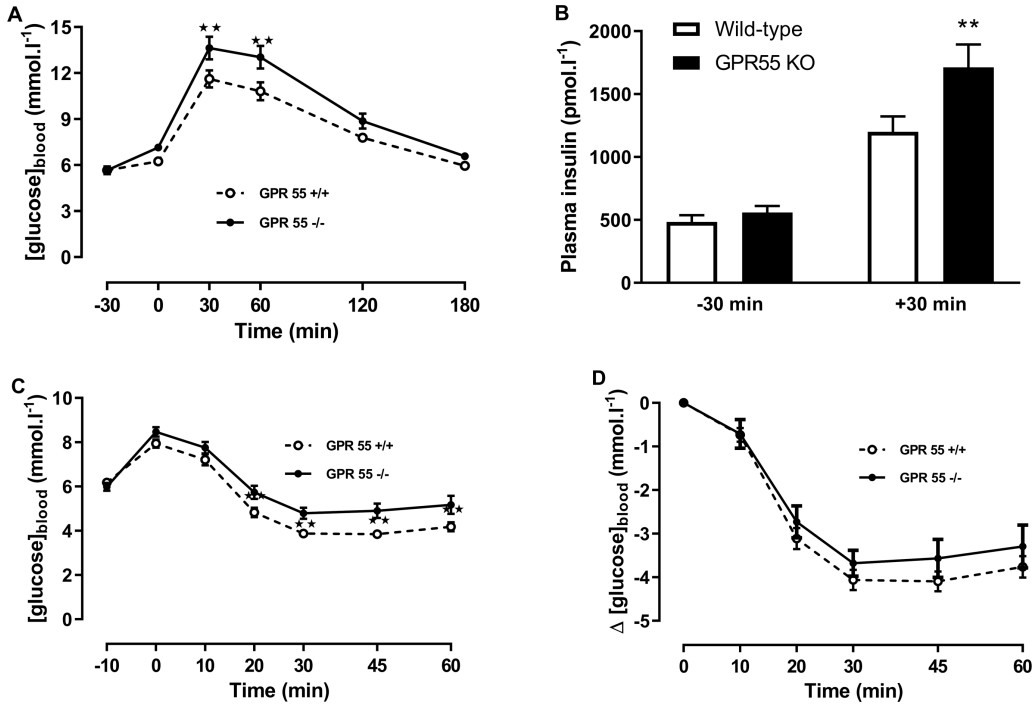

**Figure 3** **Glucose and insulin tolerance in GPR55 knockout and wild-type mice fed on a high fat diet.** Blood glucose concentration (A) and plasma insulin (B) before and after a glucose load ($t = 0$ min) at age 13 weeks after a 5 h fast, and blood glucose concentration expressed as absolute values (C) or change from $t = 0$ min (D) following an insulin load ($t = 0$ min) at age 14 weeks after a 5 h fast. Two-way ANOVA with time-matching followed by the FDR test showed that blood glucose was higher in the knockout mice at 30 and 60 min after dosing with glucose. Two-way ANOVA followed by the FDR test showed plasma insulin was higher in the knockout mice 30 after glucose. Two-way ANOVA followed by the FDR test showed higher blood glucose levels in the knockouts 20 min to 60 min following an insulin load. There was no genotype effect on the change in blood glucose following an insulin load. **$P < 0.01$.

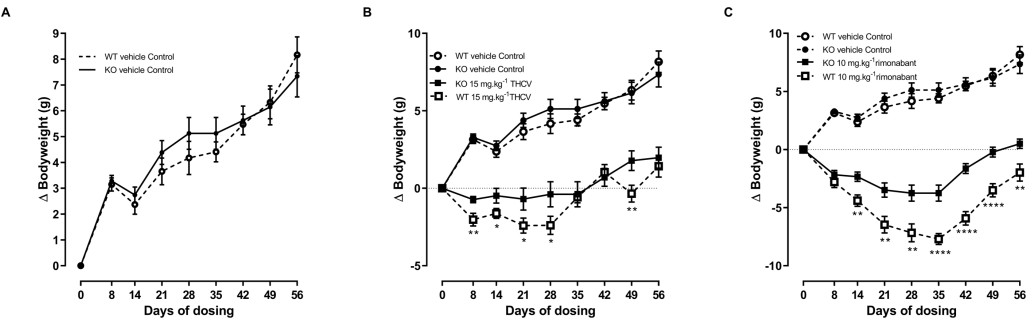

**Figure 4** **Bodyweight gain of GPR55 knockout and wild-type mice fed on a high fat diet and dosed with vehicle only (A), THCV (B) or rimonabant (C).** Vehicle only values are shown in all panels to facilitate comparisons of the effects of genotypes and drugs. Two-way ANOVA followed by the FDR test showed an overall effect of genotype on body weight gain in response to both THCV (B) and rimonabant (C). *$P, 0.05$, **$P < 0.01$, ****$P < 0.0001$.

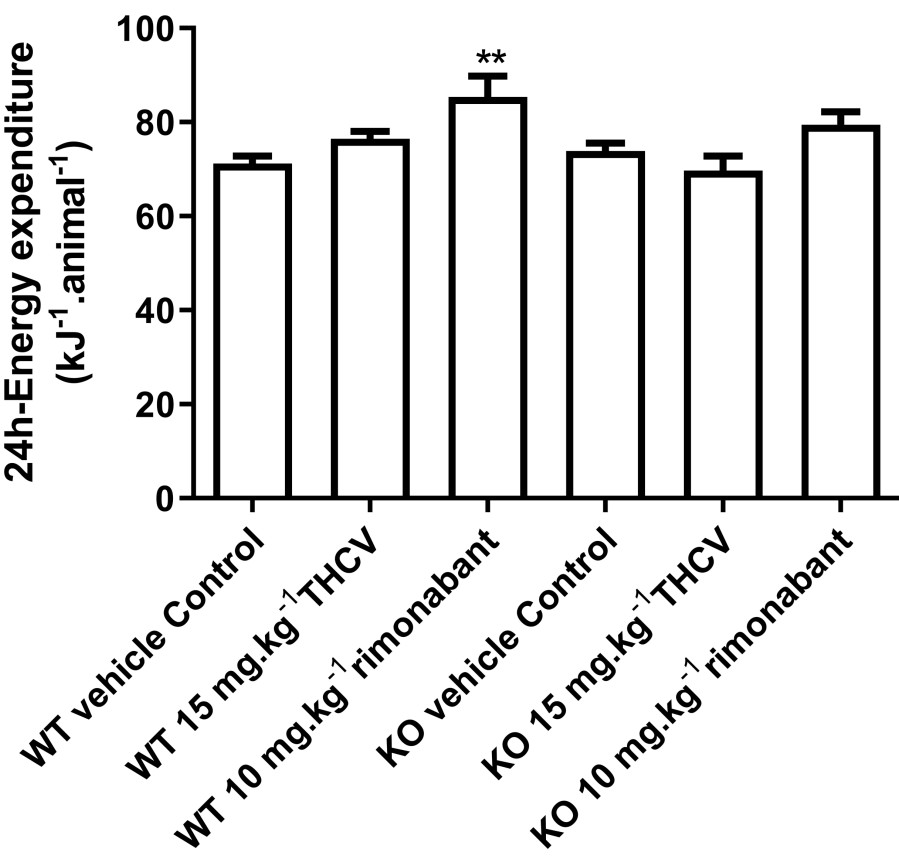

**Figure 5 Twenty-four-hour energy expenditure in mice fed on a high-fat diet and dosed with vehicle, THCV or rimonabant.** Energy expenditure measurements were performed on days 25–29 of dosing. Two-way ANOVA followed by the FDR test showed that rimonabant significantly increased energy expenditure in wild-type mice. It did not increase energy expenditure significantly in knockout mice, but there was not a significant difference between energy expenditure in rimonabant-dosed wild-type mice and rimonabant-dosed knockout mice. **$P < 0.01$.

day 56 was significant, being 33% less in knockout than wild-type mice (10.2 g different from control in wild-type mice: 6.8 g different from controls in knockout mice). Two-way ANOVA (treatment; genotype) of the weight changes on day 56 did not show an interaction between the effects of genotype and treatment ($P = 0.06$), so it cannot be claimed that GPR55 contributed more to the effect of rimonabant on weight loss than to the effect of THCV. Nor, in the absence of pharmacokinetic and concentration–response data for effects of rimonabant and THCV on GPR55- and CB1R-mediated responses in mouse tissues (such as the hypothalamus) can we attempt to explain why rimonabant caused more weight loss than THCV.

There was no effect of genotype or drug treatment on total food intake (results not shown, but are in the Data S1). Rimonabant raised energy expenditure significantly in the WT but not the KO mice on days 25–29, but there was not a statistically significant difference between its effects in WT and KO mice (Fig. 5).

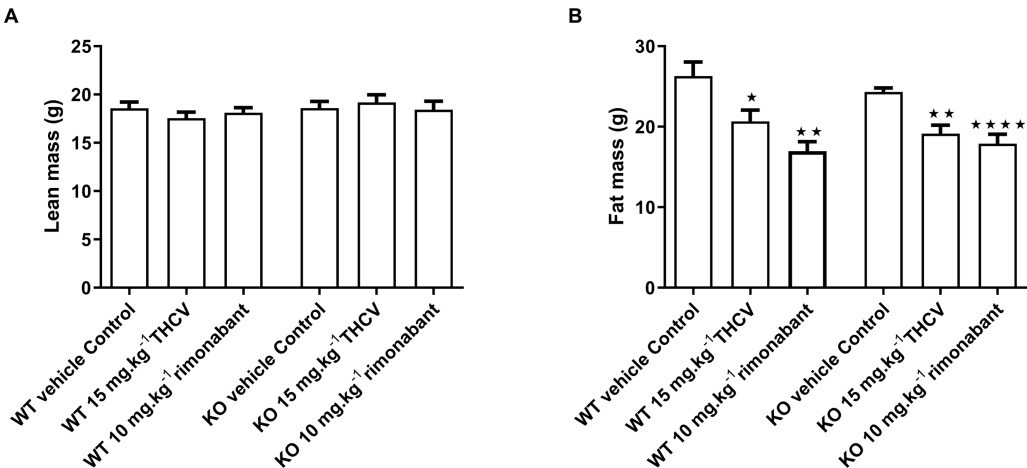

**Figure 6** **Lean and fat mass of GPR55 knockout and wild-type mice fed on a high fat diet and dosed with vehicle, THCV or rimonabant.** Body composition was measured by NMR on day 32. Two-way ANOVA followed by the FDR test showed no effect of either rimonabant or THCV on lean mass in either wild type or knockout mice (A). There was a significant effect of either THCV or rimonabant on fat mass in both wild-type and knockout mice (B). Genotype had no effect on the extent of fat mass reduction elicited by either THCV or rimonabant. *$P < 0.05$; **$P < 0.01$; ****$P < 0.0001$.

**Table 3** **Fad pad weights and locomotor activity in the main experiment.** Fat pad weights were measured at termination on day 56. Locomotor activity ($n = 4$) was measured for pairs of mice from 19:00 on day 43 to 08:00 h on day 44 when the mice were 22 weeks old and had been fed on the high fat diet for 16 weeks. Line break are given per mouse. Lights were out from 20:00 to 08:00 h. The locomotor activity was not measured for the THCV-treated mice. One-way ANOVA showed no significant effects of treatment or genotype on either fat pad weights or locomotor activity.

| | Wild-type mice | | | Knockout mice | | |
|---|---|---|---|---|---|---|
| | Control | THCV | Rimonabant | Control | THCV | Rimonabant |
| $n$ for fat pad weights | 11 | 12 | 11 | 12 | 10 | 12 |
| Epididymal fat pads (g) | $1.19 \pm 0.07$ | $1.16 \pm 0.09$ | $1.13 \pm 0.09$ | $1.29 \pm 0.09$ | $1.19 \pm 0.04$ | $1.10 \pm 0.05$ |
| Inguinal fat pads (g) | $1.10 \pm 0.15$ | $1.01 \pm 0.08$ | $0.97 \pm 0.10$ | $1.06 \pm 0.12$ | $1.04 \pm 0.13$ | $0.86 \pm 0.09$ |
| Interscapular fat pads (g) | $0.295 \pm 0.035$ | $0.380 \pm 0.076$ | $0.286 \pm 0.033$ | $0.307 \pm 0.028$ | $0.273 \pm 0.031$ | $0.217 \pm 0.027$ |
| Activity (Line breaks per 10 min) | $23.3 \pm 0.7$ | – | $21.5 \pm 0.2$ | $24.2 \pm 0.5$ | – | $22.0 \pm 1.2$ |

Locomotor activity was measured on day 45 in the control and rimonabant-treated mice only. Neither genotype nor rimonabant had any effect on total locomotor activity or its time course (results not shown, but are in the Data S1).

There was no effect of genotype or treatment on lean body mass or body fat content on day 32 (Figs. 6A and 6B). The failure to demonstrate an effect of genotype on terminal body fat content despite the effect on body weight gain being lower in knockout mice appears to be due to the initial mean body weights (and presumably body fat contents) of the knockout mice being (non-significantly) lower than those of the corresponding wild-type groups. Body fat content was less in animals treated with THCV or rimonabant than in the control group of the same genotype, but this was not reflected in significantly reduced epididymal, inguinal of interscapular fat fad weights (Table 3).

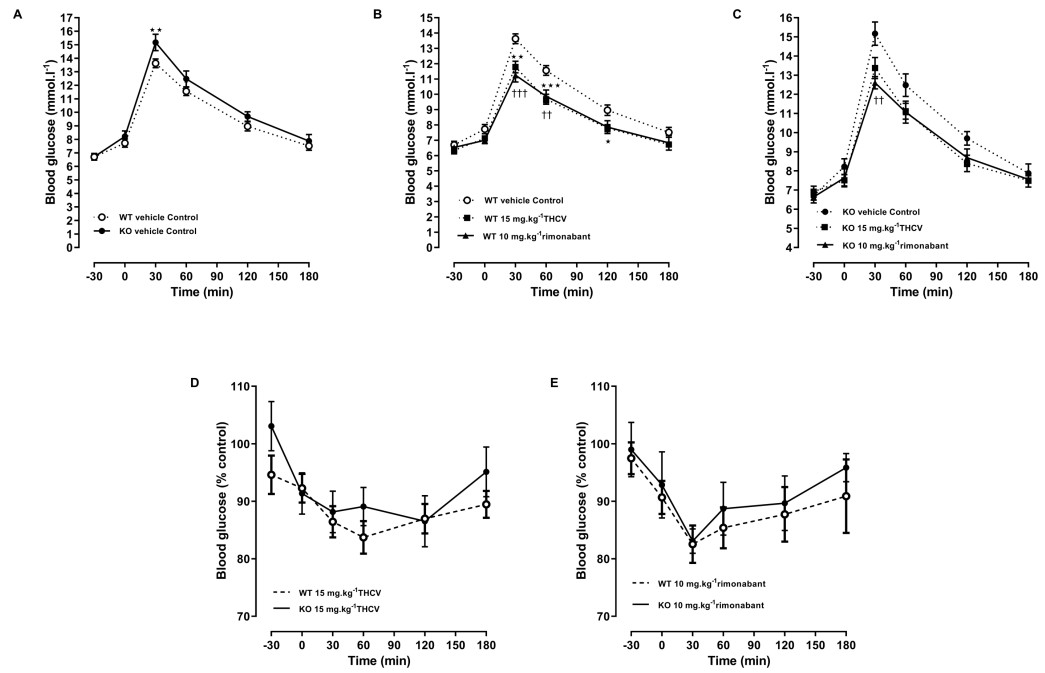

**Figure 7** **Glucose tolerance in GPR55 knockout and control mice on a high fat diet and treated with THCV or rimonabant.** Blood glucose concentrations during an oral glucose tolerance test on day 21 are expressed as wild-type vs. knockout mice (A), THCV and rimonabant-treated wild-type and knockout mice as absolute values (B and C) and relative to the respective control group in wild-type (D) and knockout mice (E). Two-way ANOVA followed by the FDR test showed that both THCV and rimonabant improved glucose tolerance in wildtype mice. Neither THCV nor rimonabant had an overall significant effect on glucose tolerance in GPR55 knockout mice, although rimonabant lowered blood glucose 30 min after glucose load. No genotype differences were observed in the relative effects of either THCV or rimonabant. *$P < 0.05$; **††$P < 0.01$; ***†††$P < 0.001$.

## Responses to THCV and rimonabant: glucose homeostasis

There was no effect of genotype on blood glucose after a 5 h fast on days 8,15 and 56 (Figs. 10A, 10B and 10C). Blood glucose was lower in the THCV-treated wild-type mice on day 15 (Fig. 10B) but not on days 8 or 56. At no time did rimonabant-treated wild-type mice or knockout mice treated with either THCV or rimonabant show reduced blood glucose (Figs. 10A, 10B and 10C).

There was also no effect of genotype on plasma insulin on days 8, 15 and 56 (Figs. 10D, 10E and 10F). On days 8 and 15 plasma insulin was lower in the THCV-treated and rimonabant-treated than in the control mice of the same genotype, although this only reached statistical significance in rimonabant-treated knockout mice on day 8 and rimonabant-treated wild-type mice on day 15. After 56 days of dosing both THCV and rimonabant reduced fasting plasma insulin in wild-type mice, but neither THCV nor rimonabant altered plasma insulin concentrations in GPR55 knockout mice (Fig. 10F). The effects of THCV and rimonabant on plasma insulin were not significantly different in wild-type or GPR55 knockout mice at any time point (Figs. 10G, 10H and 10I).

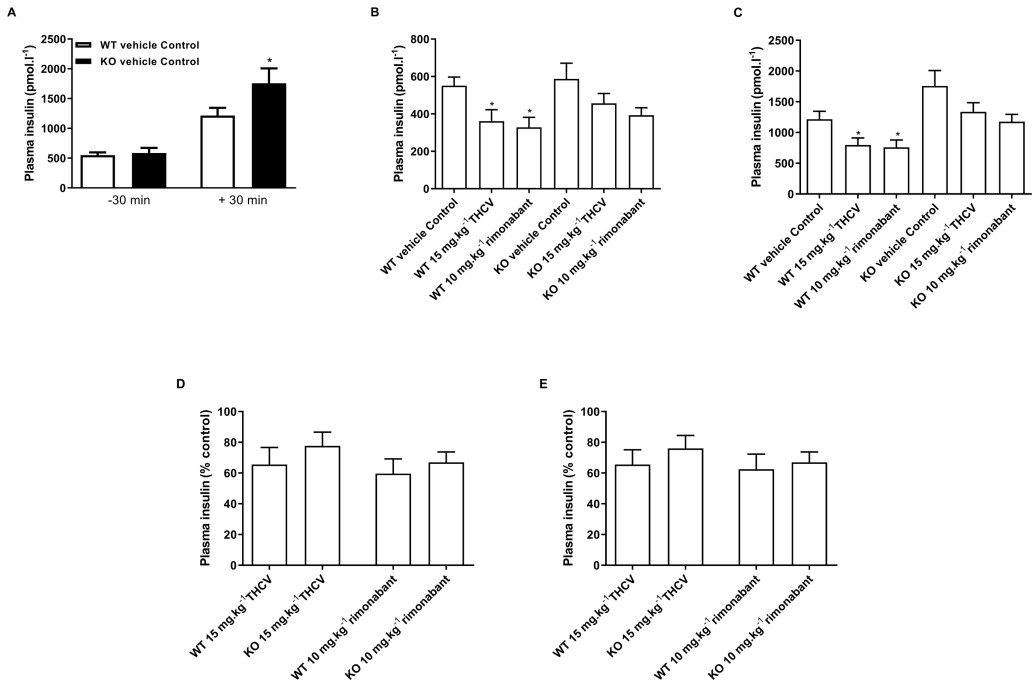

**Figure 8** **Plasma insulin concentrations during a glucose tolerance in GPR55 knockout and wild-type mice fed on a high fat diet and treated with vehicle, THCV or rimonabant.** Plasma insulin concentrations in the oral glucose tolerance test on day 21. Knockout mice had significantly higher by two-way ANOVA plasma insulin levels 30 min following an oral glucose load (A). Plasma insulin levels for THCV- and rimonabant-treated are shown 30 min before (B, absolute values; and D, relative to control group) and after (C, absolute values; and E, relative to control group) a glucose load. Two-way ANOVA followed by the FDR test showed that both THCV- and rimonabant-dosed mice had lower plasma insulin 30 before and 30 after glucose. However, the relative effects of THCV and rimonabant were not significantly different between wild-type and knockout mice. *$P < 0.05$.

An oral glucose test was conducted on day 21. THCV ($P < 0.05$) and rimonabant ($P < 0.05$) improved glucose tolerance in wild-type mice ($P < 0.05$, Fig. 7B). Neither THCV nor rimonabant had an overall significant effect on OGTT in GPR55 knockout mice, although rimonabant did significantly lower blood glucose 30 min after glucose load ($P < 0.01$), Fig. 7C. However, no genotype differences were observed in the relative effects of either THCV and rimonabant after accounting for the genotype effect on glucose tolerance in vehicle-treated mice (Figs. 7D and 7E).

THCV- and rimonabant-treated wild-type and knockout mice had lower plasma insulin concentration 30 min before (Fig. 8B) and 30 min after (Fig. 8C) a glucose load, although this only reached statistical significance in wild-type mice, although two-way ANOVA showed no interaction between the effect of treatment and genotype. Expressing the insulin concentrations relative to the respective genotype control groups also showed no significant genotype differences in the effect of either THCV or rimonabant 30 min before (Fig. 8D) or 30 min after glucose load Fig. 8E.

An insulin tolerance test was conducted on day 38. Blood glucose was lower in the THCV-treated wild-type mice than in the control wild-type mice both before and after

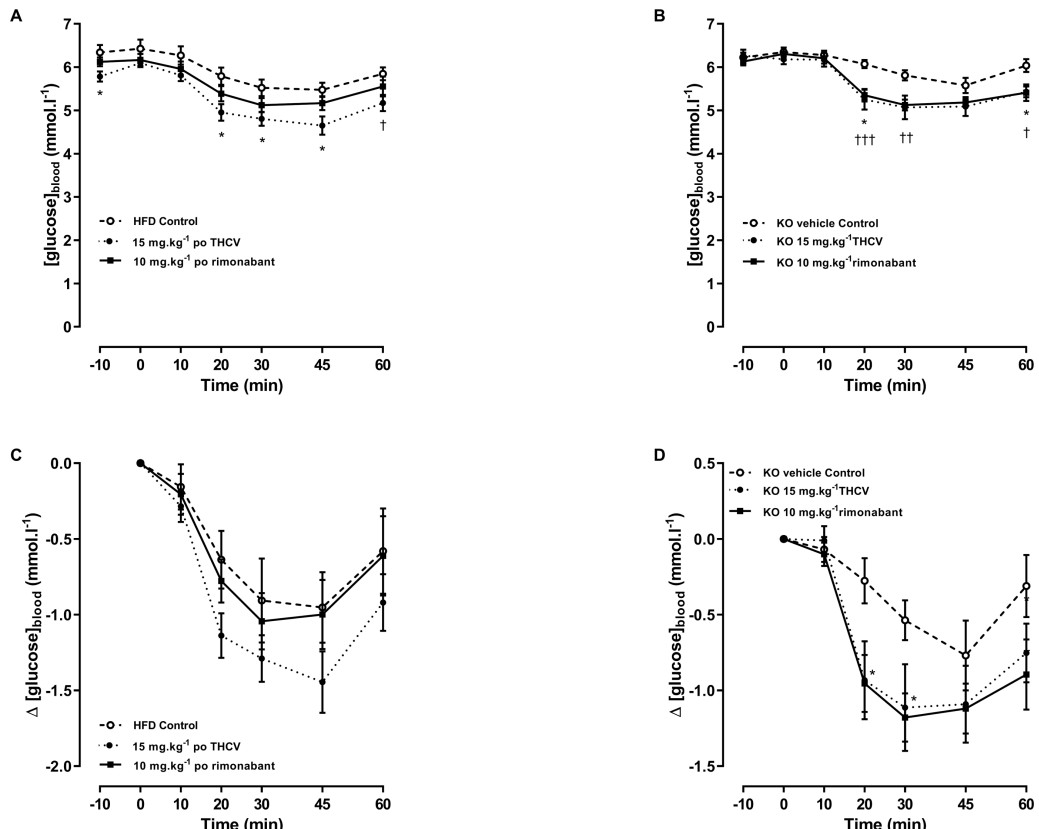

**Figure 9** **Blood glucose concentrations during an insulin tolerance test in GPR55 knockout and wild-type mice fed on a high fat diet and treated with THCV or rimonabant.** In an insulin tolerance test on day 38, two-way ANOVA followed by the FDR test showed that blood glucose was lower in the THCV-treated wild-type mice before and after administration of insulin (A). There was a small non-significant overall effect of rimonabant (A). However, there was no overall effect of either drug on the *fall* in blood glucose after giving insulin to WT mice (C). The fall in blood glucose after giving insulin to the knockout mice reached statistical significance in the THCV-treated mice (B, D) but there was no effect of genotype on the fall in blood glucose concentration with any treatment (see Fig. 8D for absolute values). *†$P < 0.05$; ††$P < 0.01$; †††$P < 0.001$.

administration of insulin. There was a small overall effect of rimonabant (Fig. 9A). However, there was no overall effect of either drug on the *fall* in blood glucose after giving insulin to WT mice (Fig. 9C).

The fall in blood glucose in the THCV-treated mice after giving insulin reached statistical significance in the knockout but not the THCV-treated mice (Fig. 9). However, these falls did not differ significantly between the genotypes, there being no effect of genotype on the fall in blood glucose concentration with either THCV or rimonabant (see Fig. 8D for absolute values).

## DISCUSSION

The present study addresses two broad questions: first, whether there are differences in energy balance and glucose homeostasis between wild-type and GPR55 knockout mice;

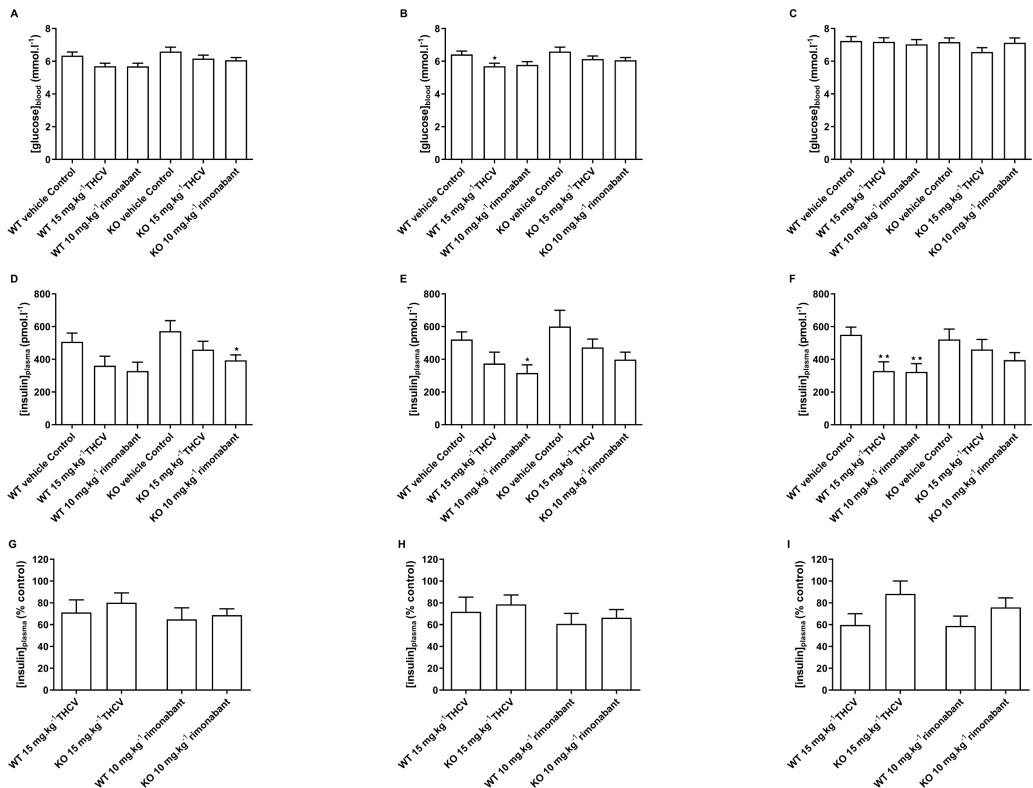

**Figure 10** **Blood glucose and plasma insulin concentrations in GPR55 knockout and wild-type mice fed on a high fat diet and treated with THCV or rimonabant.** Two-way ANOVA followed by the FDR test showed no effect of genotype on blood glucose after a 5 h fast on days 8,15 and 56 (A, B and C). On both days 8 and 15, blood glucose was lower in the THCV-treated and rimonabant-treated wild-type mice than in the control wild-type mice. Blood glucose was also lower in the rimonabant-treated knockout mice than in the control knockout mice. There was no effect of genotype on plasma insulin on days 8, 15 and 56 (D, E and F). On days 8 and 15 plasma insulin was lower in the THCV-treated and rimonabant-treated than in the control mice of the same genotype, although this only reached statistical significance in rimonabant-treated knockout mice on day 8 and rimonabant-treated wild-type mice on day 15. After 56 days of dosing both THCV and rimonabant reduced fasting plasma insulin in wild-type mice, but neither THCV nor rimonabant altered plasma insulin concentrations in GPR55 knockout mice (F). The effects of THCV and rimonabant on plasma insulin were not significantly different in wild-type or GPR55 knockout mice at any time point (G, H and I). $*P < 0.05$; $**P < 0.01$.

second whether any effects of THCV or rimonabant differ between wild-type and GPR55 knockout mice. Our main findings are that impaired glucose tolerance in GPR55 knockout mice is restricted to mice fed on a high fat diet but is not associated with increased adiposity, and that when they are fed on a high fat diet, rimonabant and THCV have less effect on weight gain in GPR55 knockout than wild-type mice. We acknowledge that for logistical and ethical reasons, including the restriction of our UK Home Office licence, we could not conduct all measurements at the ideal time. We also acknowledge that ideally, we would have conducted our experiments in both male and female mice, but it was not logistically possible to do this. We chose to use male mice because *Meadows et al. (2016)*, *Lipina et al. (2019)* and *Bjursell et al. (2016)* used male mice. If we had used female mice, any differences

between our results and theirs could be ascribed to the use of different sexes. Despite these limitations, we believe that our results have value. Other studies have the same limitations.

## Phenotypic differences between wild-type and GPR55 knockout mice

There were no differences in any aspects of energy balance (body weight, body weight change, total food consumption, daily energy expenditure, locomotor activity or body composition) between the untreated (preliminary experiment) or vehicle-treated (main experiment) wild-type and knockout mice.

Our findings differ from those of *Meadows et al. (2016)* and *Lipina et al. (2019)*, despite our mice being kindly supplied by one of the authors of the latter study. Both groups studied GPR55 knockout mice fed on chow but not on a high fat diet. The chow-fed GPR55 knockout mice had a higher mean body weight and fat content than wild-type mice. (Meadows et al. state that there is a numerical difference in body weight but acknowledge that it is not statistically significant.) The weights of some fat pads were also higher in the knockout mice. Lipina et al. reported that there was a significant reduction in lean body mass but this was expressed as a percentage of body weight and the consequence of increased fat mass. Meadows et al. found that genotype had no effect on food intake or resting metabolic rate, but spontaneous locomotor activity was lower in the knockout than the wild-type mice during the dark period.

Our results agree with those of *Bjursell et al. (2016)* in that that these investigators did not find increased body weight in the knockout mice, except for a non-significant increase in fat mass relative to body weight when the mice were fed on a 'cafeteria-fed' mice and aged 28 weeks. Our mice were killed at 15 weeks of age. These workers found *increased* locomotor activity in the knockout mice during the first 6 h of the dark period although energy expenditure was not raised at this time. During the second 6 h of the dark period, energy expenditure was no higher in their knockout than their wild-type mice and energy expenditure was depressed. We found no evidence that locomotor activity or energy expenditure was different in the knockout than the wild-type mice at any time during the dark period.

By contrast with our negative findings on energy balance, we found differences in glucose homeostasis between the untreated (preliminary experiment) or vehicle-treated (main experiment) wild-type and knockout mice in both experiments when the mice were fed on a high fat diet. The preliminary experiment found at most minor differences when the mice were fed on chow.

In both the preliminary and the main experiment, blood glucose was higher in knockout than in wild-type high fat diet-fed, untreated/vehicle-treated mice 30 and 60 min after giving glucose in an oral glucose tolerance test. The overall blood glucose level was also higher in the knockout mice. In both experiments, plasma insulin 30 min after giving glucose was higher in the knockout than the wild-type high fat diet-fed mice. These results suggest that insulin sensitivity was impaired in the knockout mice. Insulin tolerance tests failed to back this up, however. The failure to demonstrate an effect of genotype on fasting blood glucose and plasma insulin in either the preliminary or the main experiment was also unsupportive of an effect of genotype on insulin resistance. We must therefore look

beyond increased adiposity and consequent insulin resistance to account for impaired glucose tolerance in our GPR55 knockout mice.

It is feasible that impaired insulin secretion contributed to impaired glucose tolerance: plasma insulin was raised in the knockout mice after administration of glucose, but perhaps if it had been even higher glucose tolerance would have been normal. Others have reported that the GPR55 agonist O-1602 stimulated insulin secretion from wild-type but not GPR55 -/- murine islets of Langerhans (*Henstridge, Brown & Waldhoer, 2016*; *Meadows et al., 2016*) and Meadows also shows O-1602 stimulated insulin secretion and improved glucose tolerance in vivo in rats. If impaired insulin secretion is the explanation for our findings, then it seems to be exacerbated by the high fat diet.

Other workers (*Lipina et al., 2019*) have reported that GPR55 knockout mice have impaired insulin sensitivity but this may be because their mice displayed increased adiposity. In fact, although they found that blood glucose fell significantly in wild-type but not knockout mice following administration of insulin, they did not find a significant difference between genotypes. They demonstrated more clearly significant differences between genotypes in insulin signalling in isolated liver, skeletal muscle and adipose tissues. By contrast with our results, the chow-fed GPR55 knockout mice of *Meadows et al. (2016)* showed impaired insulin tolerance, but they did not exhibit impaired glucose tolerance. Meadows et al. point to raised basal insulin and a decreased response of insulin to glucose, but they did not show that these were statistically significant differences from wild-type mice. *Bjursell et al. (2016)*, who like us did not find increased adiposity in GPR55 knockout mice, have not reported studies on glucose homeostasis.

Some of the differences between our findings and those of *Meadows et al. (2016)* may be due to their conducting glucose and insulin tolerance when their mice were nine months old, whereas our chow-fed mice in the preliminary study were 16- (glucose tolerance) or 20- (insulin tolerance) weeks-old. Their mice were therefore fatter and more like our high fat-fed mice. Lipina et al. conducted their measurements when the mice were 10–22 weeks old (*Lipina et al., 2019*).

Thus, our results suggest that GPR55 interacts with insulin signalling in a more direct way than via increased fat mass. This mechanism merits further investigation. Despite many differences in details between the findings of those who have studied GPR55 knockout mice, including ourselves, we agree that GPR55 agonists might be of value in the treatment in type 2 diabetes.

## Effects of THCV and rimonabant

The beneficial effects of rimonabant on energy balance and glucose homeostasis in HFD-fed wild-type mice are well-known (*Arch, 2011*). They were reproduced in the present study.

We have previously described beneficial effects of THCV on blood glucose and plasma insulin in the fasting state and following an oral glucose load in high fat-fed obese mice (*Wargent et al., 2013a*; *Wargent et al., 2013b*). Similar results were obtained in the present study in both wild-type and GPR55 knockout mice using a dose of THCV that was towards the top of the range used in the previous study. One notable difference between the studies, however, is that in the previous study (*Wargent et al., 2013a*; *Wargent et al., 2013b*) THCV

did not affect body weight (the same was true in ob/ob mice), whereas in the present study THCV reduced body weight, weight gain and body fat content in both the wild-type and knockout mice. This was achieved without any reduction in total food intake and neither did THCV elicit a significant increase in energy expenditure. However, there was a numerical increase in energy expenditure in the wild-type (but not the knockout) mice that did not reach statistical significance, and in increase an energy expenditure *was* detected in our previous study (*Wargent et al., 2013a*; *Wargent et al., 2013b*). Energy expenditure was measured during days 25–29 only and so may not have been a reflection of the whole period of the study. Moreover, there is more variation for technical reasons in energy expenditure than in body weight and fat content and it is possible the analysis provided a false negative and it is indeed the energy expenditure that is the cause.

## Effect of genotype on the responses to THCV and rimonabant

Because oral glucose tolerance is worse in GPR55 knockout mice, there may be a greater window of opportunity for THCV or rimonabant to improve metabolism in GPR55 knockout than wildtype mice. However, if the metabolic effects of THCV or rimonabant are partly mediated by GPR55, they might be less effective in GPR55 knockout than in wild-type mice.

Nor could we detect any effect of genotype on insulin tolerance. This raises the possibility that GPR55 regulates glucose-stimulated insulin secretion. A recent study found no difference between wildtype and GPR55 knockout mice murine isolated islets of Langerhans in their responses to rimonabant (*Ruz-Maldonado et al., 2020*), which is consistent with our finding that genotype did not affect the effect of rimonabant on et al. glucose homeostasis. The authors did not discuss whether glucose-stimulated insulin secretion differed between islets from wild-type and knockout mice. THCV and rimonabant had less effect in the knockout than the wild-type mice, suggesting, that both compounds reduce body weight partly via GPR55. Based on the day-56 data, the effect of rimonabant on body weight gain was 33% less in the knockout than the wild-type mice ($P < 0.001$). The equivalent value for THCV was 19%, but these were not significantly different effects. Moreover, although the effect of genotype on the response to THCV on day 56 was not statistically significant, it was significant on a number of previous days and over all days.

The effects of rimonabant and THCV on body fat content could not be accounted for by differences in energy expenditure. We cannot, however, exclude the possibility that the latter measurements were insufficiently powered to detect a statistically significant effect. By contrast with our findings, *Bjursell et al. (2016)* found no effect of genotype on weight loss over 14 days in response to rimonabant in cafeteria-fed mice. They did not have an untreated group and they raise the possibility that the dose they used was too low, However, it was almost the same dose that we used and so we cannot explain this difference in our findings.

## CONCLUSIONS

There are varied reports on the effect of deletion of GPR55 on energy balance and glucose homeostasis in mice. Our two experiments differ from others in finding impaired glucose

tolerance in GPR55 knockout mice in the absence of any effect on body weight, body composition, locomotor activity or energy expenditure. Nor could we detect any effect on insulin tolerance. The possibility that GPR55 regulates glucose-stimulated insulin secretion merits further investigation. We also found that the reduction in weight gain elicited by THCV and rimonabant were in part mediated by GPR55.

### Funding
Funding for this study was provided by GW Research Ltd. The funders had no role in study design, data collection and analysis, decision to publish, or preparation of the manuscript.

### Grant Disclosures
The following grant information was disclosed by the authors:
GW Research Ltd.

### Competing Interests
The authors declare there are no competing interests.

### Author Contributions
- Edward T. Wargent conceived and designed the experiments, performed the experiments, analyzed the data, prepared figures and/or tables, authored or reviewed drafts of the paper, and approved the final draft.
- Malgorzata Kepczynska, Mohamed Sghaier Zaibi and David C. Hislop performed the experiments, prepared figures and/or tables, and approved the final draft.
- Jonathan R.S. Arch analyzed the data, authored or reviewed drafts of the paper, and approved the final draft.
- Claire J. Stocker conceived and designed the experiments, authored or reviewed drafts of the paper, co-ordinated the external funding, and approved the final draft.

### Animal Ethics
The following information was supplied relating to ethical approvals (i.e., approving body and any reference numbers):

University of Buckingham project licences under the UK Home Office Animals (Scientific Procedures) Act (1986) and the University of Buckingham ASPA Ethical Review Board (Animals (Scientific Procedures) Act 1986 (ASPA) Project Licence (PPL) licence number: 70/7164).

### Data Availability
The raw data is available as Supplemental File.

### Supplemental Information
Supplemental information for this article can be found online at http://dx.doi.org/10.7717/peerj.9811#supplemental-information.

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
