# Peer review of "High fat-fed GPR55 null mice display impaired glucose tolerance without concomitant changes in energy balance or insulin sensitivity but are less responsive to the effects of the cannabinoids rimonabant or Δ(9)-tetrahydrocannabivarin on weight gain"

_PeerJ, doi:10.7717/peerj.9811_

## Round 0.1 · original submission · Major Revisions

As you will see, both reviewers expressed the view the study has potential interest. However, in the light of the comments of Reviewer 1 which raise concerns regarding both methodology and interpretation, I am returning this to you for substantive revisions. I ask that you address each point raised by both reviewers, in particular the comments on assay methodology and address in particular the inconsistencies identified.

Reviewer 1 ·

Basic reporting

The written expression in this manuscript makes the content and arguments difficult to access and even appear to be logically flawed at times. Some examples where the language could be improved are:
1. The title of the manuscript, High fat-fed GPR55 null mice display impaired glucose tolerance without concomitant changes in energy balance or insulin sensitivity but are less responsive to the effects of the cannabinoids rimonabant or Δ(9)-tetrahydrocannabivarin on weight gain’, implies that GPR55 is NOT responsible for THCV mediated ‘weight gain’ whereas the authors concluded in the abstract that THCV and rimonabant ‘reduced weight gain’, and this effect was in part mediated by GPR55.
2. The statement ‘By contrast with the wildtype mice, the fall in blood glucose after giving insulin to the knockout mice reached statistical significance in the THCV-treated mice (Figures 9B, 9D) but there was no effect of genotype on the fall in blood glucose concentration with any treatment (see Figure 8D for absolute values)’ needs to be rephrased to make the argument easier to follow. Such confusing and ambiguous statements have appeared throughout the manuscript and they make the content of the manuscript difficult to comprehend.
3. In ‘The increase in energy expenditure elicited by rimonabant in knockout mice was not significant when compared to either control knockout or wild-type dosed mice’, the terms ‘control knockout’ and ‘wildtype dosed mice’ should be clarified.
4. The authors have also used the following sentence multiple times in the figure legend and it is not clear what does ‘all values are given per mouse’ actually mean.
5. It was stated in line 187-190 that ‘Food intake (results not shown), energy expenditure (Figure 5), body composition measurements (Figure 6) and fat pad weights (Table 3) also confirmed the negative results of the preliminary study’, here ‘the negative results of the preliminary study’ should be specified.

The introduction has addressed relevant background and briefly reviewed some of the key publications in the topic area. However, it was reported in a 2019 study (Ruz‐Maldonado et al), which examined the effect of Rimonabant and its analogue AM251 on insulin secretion using wildtype and GPR55 knockout mice, that Rimonabant stimulates insulin secretion via a GPR55-independent pathway. This is in direct contrast to the findings in the current manuscript that rimonabant reduced plasma insulin levels in wildtype but not GPR55 knockout mice, which suggest that the effect on plasma insulin was mediated via GPR55. The Ruz-Maldonado study also demonstrated that Rimonabant stimulates insulin secretion whereas the current study concluded that Rimonabant reduced plasma insulin levels. Although differences in in vivo and in vitro experimental settings may impact on the results, the Ruz-Maldonado study is highly relevant to the current study, should be referenced, and the discrepancies between the two studies should be addressed in this manuscript.

Figures in their current state are of very low resolution and axis labels are not visible at all in many places.

Experimental design

Despite a relatively well established role for GPR55 in regulating islet function and insulin secretion in vitro, GPR55 knockout mice do not have an apparent metabolic phenotype. It is therefore, interesting to examine whether there are any metabolic abnormities in GPR55 knockout mice under conditions of increased metabolic demand such as during diet induced obesity. The research topic of this manuscript is within the scope of the journal and the research question is of importance. However, the current manuscript has significant flaws in the design of the study, lacks accurate and sufficient details in descriptions of experimental design and methodology and failed to present some of the key results (e.g. insulin secretion/concentration and insulin content, detailed in the beginning of this review). As a result, it is very difficult to interpret the findings of this study and the study, as it stands, lacks sufficient evidence for a coherent conclusion for the research question to be drawn.

With the exception of detailed description of experimental animals, the authors offered absolutely no detail on how all of the metabolic parameters under investigation were conducted. All methodologies have been referred to previously published papers but some methods, such as the insulin tolerance tests, were in fact not described in the corresponding reference (i.e. Wargent et al 2013 was provided as a reference for insulin tolerance tests in line 137-138. However, there are two Wargent et al 2013 papers on the reference list and none of these two papers have any description on insulin tolerance test).

In terms of the experimental design, it wasn’t clear at all why various metabolic parameters were measured at particular stages of HFD feeding and comparison of these parameters, measured at different times during the HFD feeding, have been used to draw conclusions from the study. For instance, the authors speculated that ‘energy expenditure’, being ‘measured during days 25 to 29 only, may not have been a reflection of the whole period of the study’. ‘Technical reasons’, as explained by the authors, can not account for the lack of careful consideration when designing the study. Another example is that in table 2, it was stated that ‘Food intake was measured daily when the mice were between 6 and 15 weeks of age, body composition by DEXA and NMR when they were 14 and 15 weeks old respectively, energy expenditure when they were 10 to 11 weeks old, and locomotor activity when they were 13 weeks old’. The rationale for taking these measurements at these specific times should be explained. It should also be noted that under conditions of diet induced obesity, these parameters are undergoing continued change and adaptative responses towards prolonged diet modification are of great physiological relevance. As such, body composition measured when mice are 14 and 15 weeks old may not directly reflect what had happened in energy expenditure and locomotor activity (which were again measured at different time points) when mice are 10 weeks old. Furthermore, in figure 3, GTTs and ITTs were conducted ‘at age 13 weeks’ following HFD feeding since presumably age 6, whereas it was stated in Figure 9 legend that ‘there was no significant difference in the insulin tolerance curves on day 38 between the vehicle-treated knockout mice and the vehicle-treated wild-type mice (Figures 9A and 9B)’. In fact, inconsistent descriptions such as ‘Fat pad weights were measured AT TERMINATION ON DAY 56’ (table 3 legend) vs ‘our mice WERE KILLED AT 15 WEEKS OF AGE’ (line 276-277) are present throughout the manuscript; such discrepancies and/or inconsistencies in the description of key technical details make it very difficult to decipher the results obtained in this study.

Also, ‘basal’ measurements of all metabolic parameters between the wildtype and the GPR55 knockout mice should be reported to exclude any potential phenotypic differences at the start of HFD feeding.

Validity of the findings

In this manuscript entitled ‘High fat-fed GPR55 null mice display impaired glucose tolerance without concomitant changes in energy balance or insulin sensitivity but are less responsive to the effects of the cannabinoids rimonabant or Δ(9)-tetrahydrocannabivarin on weight gain’, Wargent and colleagues described their study investigating the impact of high fat diet (HFD) feeding on the metabolic parameters of the GPR55 knockout mice. Although GPR55 knockout mice fed HFD displayed impaired glucose tolerance, no significant differences in body weight, adiposity, energy balance, locomotor activity, insulin sensitivity between wildtype and GPR55 knockout mice have been seen. The authors speculated, in the discussion, that impaired insulin secretion upon GPR55 deletion could account for the impaired glucose tolerance. However, it was described in the result that ‘pancreatic insulin concentration and pancreatic total insulin content were not affected by genotype irrespective of diet’ (data from these experiments weren’t presented and it wasn’t clear how these were measured and hence the difference i.e. is the ‘pancreatic insulin concentration circulating insulin levels and if so, basal or stimulated?). In addition, Figure 3B demonstrated that plasma insulin levels in GPR55 knockout mice fed HFD was significantly higher 30 minutes after glucose injection. This is not coherent to the lack of changes in ‘insulin concentration and insulin content’ described earlier, and is not a plausible explanation to support the hypothesis that impaired insulin secretion in GPR55 knockout mice accounts for the glucose intolerance in these mice. The authors also examined whether the beneficial effects of THCV and Rimonabant on energy balance and glucose homeostasis in HFD fed wildtype mice (reported by other publications) are modulated by GPR55 deletion. Although the effect of rimonabant on reducing body weight was impaired in HFD-fed GPR55 knockout mice, the authors did not obtain conclusive evidence to explain the mechanisms involved in this.

The authors have also used flawed arguments such as:

The most likely explanation for the effect of genotype on body weight gain in response to rimonabant and THCV is that the compounds elicited a lower increase in energy expenditure in the knockout than the wild-type mice. These differences WERE NOT STATISTICALLY DIFFERENT, but the differences in the means may be sufficient to account for the differences in body fat content (line 353-356);

and

Other workers (Lipina et al., 2019) have reported that GPR55 knockout mice have impaired insulin sensitivity but this may be because their mice displayed increased adiposity. In fact, although they found that blood glucose fell significantly in wild-type but not knockout mice following administration of insulin, they did not find a significant difference between genotypes (line 303-306).

The authors attempted to conclude that GPR55 interacts with insulin signalling in a more direct way than via increased fat mass. To draw such a conclusion, it is absolutely essential to include data from experiments directly measure insulin secretion. This should both be conducted by measuring insulin secretion and insulin content from islets isolated from HFD-fed wildtype and GPR55 knockout mice, and by measuring plasma insulin before and after a glucose challenge. Indeed, the findings in figure 3B where GPR55 knockout mice had a greater insulin levels in response to glucose injection seem to be directly contradict to this conclusion.

Additional comments

Although it is a potentially interesting research question, the manuscript at its current state requires a significant revision in terms of both the study design and the way the results are interpreted and the way this study is reported.

·

Basic reporting

The figures are mostly very well presented, although a few need to be clearer – these are annotated in the text but specifically relating to figure 4 which is ineligible.

Experimental design

No comment.

Validity of the findings

The major findings of this paper are (i) that GPR55 KO mice fed a high fat diet show impaired glucose tolerance, and (ii) that the cannabinoids, THCV and rimonabant, have lesser effects in GPR55 KO animals with respect to weight gain, therefore, suggesting that the actions of these drugs are partly mediated via the GPR55. Whilst for most of the figures, statistical analyses are appropriate, there are some instances (eg. table 3), where a different statistical test should be performed.

Furthermore, the authors conclude that glucose tolerance is impaired in KO mice compared to the WT and preliminary data do support this. However, in later figures, where WT vehicle and KO vehicle are compared, these should show the same genotype differences. For example, in figure 9, if you directly compare the WT vehicle and KO vehicle curves, do they show the same differences as in figure 2 - this is important, particularly given that this is one of the major conclusions and the title of the manuscript. This also relates to a genotype comparison in figure 8 panel C.

Additional comments

The manuscript is generally well written and with clear aims/objectives and conclusions. The main conclusions of this study are that (i) GPR55 KO mice fed a high fat diet show impaired glucose tolerance, and (ii) that the cannabinoids, THCV and rimonabant, have lesser effects in GPR55 KO animals with respect to weight gain, therefore suggesting that the actions of these drugs are partly mediated via the GPR55.

Overall, the conclusions of the study are supported by the data presented. However, I have a few general points which should be addressed:
1. What is the rationale behind using only male mice? It would be useful to state this in the methods. Have you conducted any similar analyses in female mice, and do they show similar genotype differences?
2. Direct comparisons between the effects of the two treatments, THCV and rimonabant, cannot really be drawn (e.g. when discussing figure 4) without PK/PD data showing relative levels of exposure to each compound. Furthermore, data to show the efficacy of each drug at the GPR55 in vitro would be useful. This data may already be published, but it would enhance the current manuscript to compare the action of these two ligands at the GPR55 in vitro. This leads me to the question - is rimonabant having a greater effect in this dataset because it is more efficacious at the GPR55 or is the exposure higher to this ligand in vivo? The introduction and/or discussion should consider this point, and perhaps the statements around a greater effect of rimonabant on weight gain could be softened.
3. Generally, it would be useful to ensure that statistical analyses and results of such analyses are outlined in every figure legend. There are some inconsistencies in the way that statistical comparisons are described.
4. When comparing treatment effects and genotype effects, such as in table 3, a one-way ANOVA is not appropriate. These data should be analysed using a two-way ANOVA.
5. It would markedly improve the manuscript if bar graphs were replaced with scatter charts so that the variability of data within each experiment is displayed.
6. In all figures, are data displayed as means -/+ SEM or SD? This needs to be stated in the figure legends, as this is important when considering the statistical analyses.
7. In figures 7 and 8, are there any effects of the two drugs on the KO? The trend seems to be similar for the WT, but this is not displayed as a significant effect? In figure 8 panel C, the comparison between WT vehicle and KO vehicle should be significant as this would be the same as in panel A – is this the case?
8. For figure 9, if the authors were to directly compare the WT and KO control curves – would these show similar differences to earlier figures? I.e. do they support earlier conclusions and in fact the title of the manuscript?

---

## Round 0.2 · Minor Revisions

As you will see, one of the reviewers has some remaining concerns which have not been addressed in your more recent revision. It is important to address these please, and re-submit with a clearly marked-up revised version. I would not anticipate that this will present any significant difficulties for you.

·

Basic reporting

In my previous review, I suggested that scatter charts could replace bar charts to display the variation between individual mice and I still think that this would markedly improve the presentation of the data.

In all figures, are data displayed as means -/+ SEM or SD? This needs to be stated in the figure legends, as this is important when considering the statistical analyses.

Experimental design

Previously I questioned the rationale for the sole use of male mice. I wonder if it is possible to state the reasons why comparisons have not been drawn in female mice as well.

Validity of the findings

In my previous review, I raised the following points:

1. It is difficult to discuss a comparison between the efficacy of two treatments, THCV and rimonabant, without consideration of the relative exposure levels. Is rimonabant having a greater effect in this dataset because it is more efficacious at the GPR55 or is the exposure higher to this ligand in vivo. This should be addressed in the discussion.

2. What is the explanation for why the WT vehicle control and KO vehicle control plasma insulin levels (Figure 8C) are not statistically different. Is this the same data as displayed in Figure 8A? Where the KO vehicle control is statistically higher than the WT vehicle control 30 minutes after the glucose load?

3. In figure 9 have the WT and KO control curves been directly compared and do any genotype differences here support the earlier conclusions in this manuscript?

Additional comments

The majority of my previous comments have been addressed, but there remain a few points outlined above that should also be considered.

---

## Round 0.3 · accepted · Accept

Thank you for addressing the issues raised at the second round of review.